A comparison of non-surgical methods for sexing young gopher tortoises (Gopherus polyphemus)

Loope Kevin J. kjloope@vt.edu 1 2
Rostal David C. 1
Walden M.A. 3
Shoemaker Kevin T. 3
Hunter Elizabeth A. 1 2 4
1 Department of Biology, Georgia Southern University , Statesboro , GA , United States of America
2 Fish and Wildlife Conservation, Virginia Polytechnic Institute and State University (Virginia Tech) , Blacksburg , VA , United States of America
3 Department of Natural Resources and Environmental Science, University of Nevada—Reno , Reno , NV , United States of America
4 Virginia Cooperative Fish and Wildlife Research Unit, U.S. Geological Survey , Blacksburg , VA , United States of America
Kistler Whitney
Electronic publication date: 2022 Jun 14
Publication date: 2022
Volume: 10
Electronic Location ID: e13599
Received 2022 Mar 10; Accepted 2022 May 26
Copyright year: 2022
Copyright holder: Loope et al.
License: This is an open access article, free of all copyright, made available under the Creative Commons Public Domain Dedication. This work may be freely reproduced, distributed, transmitted, modified, built upon, or otherwise used by anyone for any lawful purpose.
License URL: https://creativecommons.org/publicdomain/zero/1.0/

Keywords: Sex determination, Testosterone, Chelonians, Tortoises, Follicle-stimulating hormone, Geometric morphometrics, Sex hormones

Funding: Georgia Southern University Research Development Fund, and the Department of Defense Strategic Environmental Research and Development Program RC18-C1-1103 This work was supported by funding from the Georgia Southern University Research Development Fund, and the Department of Defense Strategic Environmental Research and Development Program (RC18-C1-1103). The funders had no role in study design, data collection and analysis, decision to publish, or preparation of the manuscript.

==============================
Many turtle species have temperature-dependent sex determination (TSD), raising the prospect that climate change could impact population dynamics by altering sex ratios. Understanding how climate change will affect populations of animals with TSD requires a reliable and minimally invasive method of identifying the sexes of young individuals. This determination is challenging in many turtles, which often lack conspicuous external sexual dimorphism until years after hatching. Here, we explore four alternatives for sexing three age classes of captive-reared young gopher tortoises (Gopherus polyphemus), a terrestrial turtle of conservation concern native to the southeastern United States: (1) naive testosterone levels, (2) testosterone levels following a follicle stimulating hormone (FSH) challenge, (3) linear morphological measurements, and (4) geometric morphometrics. Unlike some other turtle species, male and female neonatal gopher tortoises have overlapping naive testosterone concentration distributions, justifying more complicated methods. We found that sex of neonates (<7 days old) is best predicted by a “random forest” machine learning model with naive testosterone levels and morphological measurements (8% out-of-bag error). Sex of hatchlings (4–8 months old) was predicted with 11% error using a simple threshold on naive testosterone levels, or with 4% error using a simple threshold on post-FSH testosterone levels. Sex of juveniles (approximately 3.5 years old) was perfectly predicted using a simple threshold on naive testosterone levels. Sexing hatchlings at >4 months of age is the easiest and most reliable non-surgical method for sex identification. Given access to a rearing facility and equipment to perform hormone assays, these methods have the potential to supplant laparoscopic surgery as the method of choice for sexing young gopher tortoises.

Introduction

Unlike the chromosomal sex-determination system of most vertebrates, sex in many reptile species is determined by the temperature experienced during a critical window of embryonic development. Temperature-dependent sex determination (TSD) has evolved from genetic sex determination at least 11 times, and is found in crocodilians, tuatara, numerous turtles, and some lizards (Cornejo-Páramo et al., 2020), and may have evolved because the two sexes have different thermal optima during development (Charnov & Bull, 1977). Despite its deep evolutionary history, TSD may make populations vulnerable to rapid environmental shifts, as is occurring with human-induced climate change (Mitchell & Janzen, 2010). Rapid changes in nest thermal environments could quickly produce extremely biased sex ratios (Jensen et al., 2018), causing a rapid reduction in effective population size or even population collapse before evolutionary responses to this novel selection pressure could occur (Mitchell & Janzen, 2010). The degree to which physiological and behavioral processes could cushion populations from rapid environmental change is unknown and likely to be species- and context-specific. For example, females could respond to warming by plastically adjusting oviposition times, nest sites, or nest characteristics such as depth in ways that ameliorate the effects of warming on nest temperatures, and thus avoid drastic swings in sex ratio at the population level. Alternatively, it may be that populations are locally and rigidly adapted to environmental conditions, and that ambient warming could lead directly to nest warming and extreme population sex ratios, as appears to be occurring in at least one sea turtle population (Jensen et al., 2018). Identifying methods that reliably and non-destructively assess neonate sex may be critical for understanding the population trajectories of imperiled species with TSD into the future.

There are numerous approaches to sexing young turtles. Hatchlings vary in the degree of sex differentiation in external characteristics, but many species appear externally monomorphic, making sex identification challenging. Dissection of sacrificed individuals and inspection or histological study of the gonads is the gold standard for identifying sex (Wibbels, Bull & Crews, 1991; Spotila et al., 1994; Wibbels, Rostal & Byles, 1998; Wyneken et al., 2007; Lazar et al., 2008; LeBlanc et al., 2012; King et al., 2013). However, to avoid sacrificing young turtles, non-lethal methods have also been developed. The visual inspection of gonads in live animals via laparoscopic surgery is one approach, and this procedure typically results in high accuracy of sex identification and high survival rates (Rostal et al., 1994; Spotila et al., 1994; Wyneken et al., 2007). However, laparoscopy requires specialized surgical skills, equipment and rearing facilities (neonates must be reared until the yolk is completely absorbed, often taking months), and still presents some risk of mortality and other adverse consequences. Thus, less invasive techniques have been developed in a variety of species. In a population of olive ridley sea turtles (Lepidochelys olivacea), simple combinations of linear morphological measurements are sufficient to distinguish sexes with 95% confidence (Michel-Morfin, Muñoz & Rodríguez, 2001), and a similar approach has been suggested for juvenile gopher tortoises (Burke et al., 1994). A more sophisticated method was developed for Podocnemis expansa and Chrysemys picta to detect subtle shape differences using geometric morphometrics (Valenzuela et al., 2004), and has been also applied successfully in the snapping turtle Chelydra serpentina (Ceballos & Valenzuela, 2011) and another river turtle, Podocnemis lewyana (Gómez-Saldarriaga, Valenzuela & Ceballos, 2016). Circulating plasma hormone concentrations have also been shown to successfully differentiate sex in young turtles. Two sea turtle species, lacking apparent morphological separation of sexes at hatching, have been sexed using estradiol:testosterone ratios in blood (Xia et al., 2011) or amniotic fluid (Gross et al., 1995; Xia et al., 2011) with 96% accuracy. Baseline plasma testosterone is sufficient to sex captive-reared juveniles of the Mojave desert tortoise, Gopherus agassizii (Rostal et al., 1994) and wild juvenile green sea turtles, Chelonia mydas (Allen et al., 2015), and in Arrau (P. expansa) and snapping turtles (Chelydra serpentina), testosterone levels separated sexes after a follicle stimulating hormone (FSH) challenge disproportionately elevated male testosterone levels (Lance, Valenzuela & Von Hildebrand, 1992; Ceballos & Valenzuela, 2011). These non-surgical methods vary in their accuracy, cost, age at which the procedure can be performed, and degree of expertise and equipment required, with morphological methods being more efficient and less invasive than hormone methods, although vulnerable to potential differences in morphology between populations and in some cases less accurate.

Tortoises in the genus Gopherus are found in arid habitats throughout the southern parts of North America (Nussear & Tuberville, 2014). The gopher tortoise (G. polyphemus) has declined due to habitat loss and degradation, and overconsumption (Berry & Aresco, 2014), and is federally protected in the western part of its range in Louisiana and Mississippi, and state-protected throughout the rest of its range in the southeastern coastal plain of Alabama, Georgia, Florida and South Carolina. Warming temperatures are associated with higher fecundity in gopher tortoise females in one northern population (Hunter et al., 2021), but to date no studies have addressed the effects of climate change on primary sex ratios in gopher tortoise populations. A major barrier to such studies is the lack of simple, accurate sexing protocols for young tortoises, given that secondary sexual characteristics are not apparent until the age of at least 10 years (Mushinsky, Wilson & McCoy, 1994). Currently, the sex of young gopher tortoises is determined by laparoscopy after the age of ∼6–12 months, a procedure requiring specialized equipment and access to a trained surgeon and a rearing facility. Captive rearing of hatchlings is necessary to obtain tortoises of sufficient size for the endoscope, and to allow for absorption of the yolk (Rostal et al., 1994; Rostal & Wibbels, 2014). The ideal method would be non-invasive or only require a blood draw, and be effective in neonates, allowing release immediately after hatching, rather than an extended period of rearing prior to the assay. A non-surgical sexing method for young gopher tortoises could allow a far greater number of gopher tortoise researchers to assess hatchling sex, facilitating range-wide assessment of the effects of climate change on population sex ratios.

Based on a survey of methods developed for various chelonians, we identified four potential approaches to determining sex in young gopher tortoises. First, baseline plasma testosterone may be higher in males than females, which could allow accurate sexing (Rostal et al., 1994). In the related Mojave desert tortoise, testosterone distinguished males and females for both juveniles (57–90 mm SCL) and immatures (136–190 mm), though neonate levels were not assessed (Rostal et al., 1994). Second, if baseline testosterone does not separate sexes, male testosterone may be differentially elevated by administering exogenous follicle stimulating hormone (Lance, Valenzuela & Von Hildebrand, 1992). Third, a simple linear combination of gross morphological measurements may distinguish males from females (Burke et al., 1994). Fourth, subtle shape differences between sexes may be detected using geometric morphometrics (Valenzuela et al., 2004). We evaluated potential sexing methods for three age classes of young gopher tortoises maintained in the laboratory post-hatching, using laparoscopic surgery (Rostal et al., 1994) to validate the true sex of assayed individuals.

Materials and Methods

Study animals

We tested several sexing methods on three age classes of immature gopher tortoises. “Neonates” were assessed within 1 week of hatching, “hatchlings” were sampled at ∼4–8 months of age (mean carapace length: 66.2 mm, range: 58–76 mm; for definitions of age/size classes of gopher tortoises, see Ashton & Ashton, 2008), and “juveniles” were sampled at approximately 45 months of age (mean carapace length: 145 mm, range: 101–170 mm). To obtain young tortoises, eggs were incubated in natural nests in the ground at Fort Stewart Army Reserve and George L. Smith State Park in southeastern Georgia, USA (Rostal & Jones, 2002; Hunter & Rostal, 2021). After the critical sex determination period (Rostal & Wibbels, 2014) but prior to hatching, we moved eggs to incubators where they were maintained at 29.9 °C until hatching. Eggs of assayed neonates and hatchlings were collected in 2018–2020, and eggs of juveniles were collected in 2015. Captive tortoises were reared in groups of 1–10 in plastic bins with plastic or ceramic hides. They were fed daily with a diet of mixed greens and provided with water twice weekly. All procedures using these animals were registered with and approved by the Institutional Animal Care and Use Committee of Georgia Southern University (#I18023) and the Georgia Department of Natural Resources (#1000545889 and #1000838720). The protocol entailed euthanasia by a registered veterinarian in the case of damage to internal organs during surgery. No euthanasia was required. Surviving animals were released after at least 1 month of recovery in captivity.

Quantitative testosterone hormone assay for Gopherus polyphemus

We collected approximately 100 µl of blood from the subcarapacial sinus (neonates and hatchlings) or the brachial vein (juveniles). Blood was quickly transferred into lithium heparin-treated glass capillaries (70 mm; Fisher Biosciences1) and immediately centrifuged for 3 min at 6,000 rpm. Plasma was stored at −20 °C until extraction. We extracted testosterone from 30–50 ul of plasma by combining with one mL of anhydrous diethyl ether (Acros Organics) in a glass test tube, then vortexing for 20 seconds. The tube was then held against a block of dry ice until the aqueous layer was frozen (20 seconds), and the organic layer was then decanted into a clean test tube. Ether was evaporated in a fume hood overnight, and samples were then ready for enzyme-linked immunosorbent assay (ELISA). Samples were re-eluted the next morning in ELISA buffer (Item No. 582701, Cayman Chemical, Ann Arbor, MI) and vortexed for 20 seconds. We then performed ELISAs using the Cayman Chemical Testosterone ELISA kit following manufacturer instructions, running all samples in duplicate.

We validated that the ELISA detected the same target as the standard by checking for parallelism between an 8-point, twofold serial dilution of the manufacturer-provided testosterone standard and a 10-point, twofold serial dilution of an extracted pool of tortoise plasma (20 adult females from the same population). We calculated extraction efficiency and checked for matrix interference by spiking 25–30 µl aliquots of juvenile female plasma (n = 6) with 10 pg of testosterone (5 µl of a 1:100 dilution of the Cayman supplied standard). Samples were then extracted and run in duplicate as described above. Extraction efficiency was calculated for each sample using the formula E = (Mspiked − Munspiked)/S where M is the measured amount of testosterone in the spiked or unspiked sample, and S is the known amount of testosterone added in the spike (in pg).

Samples were run in duplicate with up to 32 samples per plate, as well as reference standards. The reference standard was a dilution of pooled plasma from 20 adult female G. polyphemus collected in 2007 and stored at −20 °C. We used these standards to calculate between- and within-plate percent coefficients of variation (% CV).

Laparoscopy to determine sex

Evaluating non-invasive sexing methods requires knowing the sex of subjects to determine the accuracy of each approach. We used laparoscopy to determine sex for tortoises when they were 6–12 months old, using the method described by Rostal et al. (1994). However, we modified this protocol by administering lidocaine as a local anesthetic (Hernandez-Divers, Stahl & Farrell, 2009; Rakotonanahary, Kuchling & Routh, 2015; Emmel et al., 2021), and sealing the incision with Surgi-Lock 2oc (Meridian) glue following the procedure. Tortoises were monitored individually for 24–48 hours after surgery until they were observed feeding and behaving normally, then returned to their housing. Surgeries were performed at least 1 week after each individual’s last blood draw to ensure they had fully recovered before the operation. Laparoscopies were performed when tortoises were at least 50 g body mass, and after at least 6 months post-hatching to ensure the yolk had been completely absorbed. For juveniles hatched in 2015, laparoscopies occurred at approximately 1 year of age. The same person (D. Rostal) conducted laparoscopies in all cases. The surgeon was blind to individual sex, and all associated data collection (testosterone, morphology) was done blind to the sex determined by laparoscopy.

Sexing approach 1: baseline plasma testosterone

Using the validated ELISA method described above, we quantified plasma testosterone in neonates, hatchlings and juveniles (see sample sizes in Table 1). Samples were run in haphazard order by individual ID (not determined by sex). Sample sizes were determined by the number of available tortoises large enough to laparoscopy (>50 g) in each year. No animals with sex determined by laparoscopy were excluded.

Table 1 Sample sizes for naive and FSH-challenged testosterone samples by age of gopher tortoise (Gopherus polyphemus) collected from wild nests in Georgia in 2018–20.

Age	Treatment	Females	Males	
Neonate (<7 days)	Naive	44 (2020)	18 (2020)	
Neonate (<7 days)	FSH-challenged	16 (2019)	32 (2019)	
Hatchling (4–8 mo)	Naive	66 (2020) & 13 (2018)	25 (2020) & 9 (2018)	
Hatchling (4–8 mo)	FSH-challenged	65 (2020) & 14 (2018)	23 (2020) & 11 (2018)	
Juvenile (∼3.5 yr)	Naive	11 (2015)	12 (2015)	
Notes.

Parenthetical values are year of hatching. In 2018 and 2020, many of the same individuals were assayed before and after FSH challenge.

Sexing approach 2: testosterone following follicle stimulating hormone challenge

We injected tortoises with FSH to see if FSH differentially elevates male testosterone, allowing the separation of males and females based on a subsequent plasma testosterone assay. Juveniles were not challenged with FSH, as naive testosterone levels clearly distinguished the two sexes (see results). In 2018, we tested three doses of FSH: young juveniles were injected with either 0.1 (n = 12), 0.05 (n = 12), or 0.01 (n = 15) units of porcine FSH (MP Biomedicals 0210172750) in 100 ul sterile saline. Injections were performed intracoelomically (Lance, Valenzuela & Von Hildebrand, 1992). If the individual had been sampled for baseline plasma testosterone, the injection was performed at least 6 days after the earlier blood draw. Four to six hours after injection, ∼100 µl of blood was drawn from the subcarapacial sinus. Samples were processed, stored and testosterone was quantified using the methods described above. For a subset of individuals (n = 15), we repeated this sampling six days after injection to determine the longevity of the effects of FSH treatment. A subset (n = 11 males and n = 14 females) were later laparoscopied to determine sex.

To verify that observed effects were not due to the stress of handling and injection rather than FSH itself, four hatchlings (later determined to be two males and two females) were injected with 100 ul of sterile saline and sampled 4 hours later. These individuals were later laparoscopied, and 1 month following surgery treated with 0.01 units of FSH and sampled (included in the sample size reported above).

After establishing the lowest effective dose, we similarly administered 0.01 units of FSH by injection (followed by blood draws and testosterone quantification) for neonates in 2019 and hatchlings in 2020 (Table 1). These animals were then maintained in the lab and sexed via laparoscopy. For subsequent analysis of post-FSH hatchlings, we only used data from 2020 individuals, when all individuals were administered the same dose of FSH.

Testosterone analysis

All analyses were performed in R v4.1.0 (R Core Team, 2021). We estimated several thresholds to use in distinguishing males and females for each testosterone dataset (sexing approaches 1 and 2). First, we estimated the threshold that minimized the total percentage of erroneous sex assignments (herafter Tm). This total percentage is calculated by estimating the percentage of erroneously assigned males and the percentage of erroneously assigned females and summing them; this avoids bias that would occur if we used total numbers of erroneous assignments for datasets with unequal sex ratios. Then, since Tm could be biased toward one type of error (i.e, mistaking males for females or vice versa), resulting in a bias in the overall estimate of sex ratio, we estimated the threshold associated with an equal rate of misidentifying true females and true males. This was calculated by generating a sequence of possible testosterone values from the minimum to the maximum observed values by 0.1 units, then estimating the error rate (the fraction of observed females with testosterone values above the threshold, or the fraction of males with values below the threshold) for each sex for each possible testosterone value. We then selected the testosterone value that resulted in the minimum difference in male and female error rates. Finally, given the substantial overlap in testosterone concentration distributions between sexes for some datasets, we sought a method to filter ambiguous individuals and only assign sexes in individuals with testosterone values outside of the ambiguous range. To do this, we calculated the threshold below which the probability of an individual being female was 90% (TF90), and the threshold above which the probability of an individual being male was 90% (TM90). To calculate the former, we used Bayes’ Theorem: (1) Psex=“female”|T<x0=PT<x0|sex=“female” ∗ Psex=“female”PT<x0.

When we assume a given individual has equal probability of being either sex, this reduces to: (2) PT<x0|sex=“female”PT<x0|sex=“female”+PT<x0|sex=“male”,

which can be estimated from our datasets for each sex. To calculate TF90, we generated a sequence of testosterone values from the minimum to the maximum observed testosterone by 0.1 units and used the above formula to calculate corresponding probability of being female given observed concentrations of testosterone less than a threshold value. We then fit a smoothed spline to the probabilities and from this curve estimated the testosterone value below which an individual is 90% likely to be female. We calculated T90M in the same way, as well as analogous thresholds for 80% and 95% likelihood for each sex. This approach allowed us to avoid fitting distributions to our testosterone datasets, as the male testosterone values were in some cases not well fit by lognormal or gamma distributions. To validate error rates from the application of TE and TM thresholds, we estimated error rates using leave-one-out cross-validation of each sample, generating errors that are comparable to out-of-bag (OOB) error rates from random forest models (see below). To quantify sampling variation, we computed 95% quantiles from bootstrapped sampling distributions (n = 1, 000 replicates) for all thresholds and error rates.

To test for a differential effect of FSH challenge on testosterone levels for males and females, we used permutation t-tests (perm.t.test(), Kohl, 2020) on the absolute and percentage change in testosterone for 23 male and 65 female hatchlings for which we had both pre- and post-FSH testosterone levels.

Sexing approach 3: Morphological measurements

For each neonate used in the testosterone experiments, we took five morphological measurements (midline carapace length (MCL), shell height (SH), carapace width (CW), minimum and maximum plastron length (PMIN and PMAX)) in the first 1–2 weeks after hatching in 2018, 2019 and 2020. Hatchlings were measured at 7.5–8.5 months post-hatching (n = 78 in 2020), and for these individuals we also measured anal width (AW), anal notch depth (AN), gular width (GW) and tail length (TL) (Fig. 1; Burke et al., 1994). Juveniles were not measured. Sex was determined by laparoscopy. We calculated several derived variables based on these measurements that were suggested to be sex-informative in an earlier study (VOL = MCL * SH * CW, MCL.V = MCL/VOL, SH.V = SH/VOL, CW.V = CW/VOL, PMIN.V = PMIN/VOL, PMAX.V = PMAX/VOL, CW.MCL = CW/MCL, CW.SH = CW/SH; Burke et al., 1994). All raw measurements, as well as the composite measurements, were included in a random forest model (Liaw & Wiener, 2002) to see if they could be used to classify neonate sex, using mtry = 5 and ntrees = 105. Because sexes were not equally represented in some datasets, we specified sampsize = c(x,x) to downsample the larger class such that class frequencies were equal, where x is the number of samples in the rarer class. We created models using only morphological data for the full dataset, and using morphological data and testosterone data for two subsets of neonates, those with an FSH challenge (2019) and those without (2020), and compared these to models with only testosterone included (in these cases, mtry =1), to see if morphological data improved sex identification by testosterone level alone. We did not use linear discriminant function analysis (as done by Burke et al., 1994) due to high multicollinearity among predictors.

Figure 1 Morphological measurements from young gopher tortoises (Gopherus polyphemus) collected from wild nests in Georgia in 2018–20.

Full names of each measurement are given in the text. Shell height (SH) not depicted. Anal notch (AN), anal width (AW) and gular width (GW) only measured in hatchlings. Tail length (only measured in hatchlings) not shown.

Sexing approach 4: Geometric morphometrics

To test for subtle differences in shape between sexes, we used a geometric morphometric approach to quantify plastron and carapace morphology in neonates (n = 21 females; n = 15 males). We followed the procedures detailed by Valenzuela et al. (2004) to quantify morphometric differences among individuals. First, we took standardized photographs of the top of the carapace (top) and plastron (bottom) with one measurement (suture intersection measurement, SIM) per photograph to provide a scaling factor that was most in-line with the plane of important shell landmarks (Fig. S1). We marked the x,y coordinates of standardized landmarks (scute intersection points) and aligned them using TpsDig software (Rohlf, 2018). We delineated 28 top and 14 bottom landmarks (Fig. S1). We extracted partial warp scores from the thin-plate spline using TPSRelw software (Rohlf, 2018) which capture the variation in shape among individuals (n = 51 top and 24 bottom partial warp scores). We created a random forest model following the process for morphological measurements described above using all partial warp scores.

Results

Testosterone ELISA validation

The Cayman ELISA assay demonstrated good parallelism between serially diluted tortoise plasma and serially diluted standards in the range of 10–90% bound/maximum bound tracer (B/B0), the range in which male and female testosterone concentration distributions overlapped and thus the range where accurate testosterone concentrations were most important (Fig. S2). This parallelism demonstrated that the kit was likely detecting testosterone in the tortoise plasma, and that the plasma lacked matrix interference that could bias quantification. Estimated extraction efficiency was high, averaging 94.1% (+/-16% SD) for six spiked samples. Between-plate % CV for plates with 2018 samples was 7.6%, with a within-plate % CV of 9.2%. Between- and within-plate % CVs for the remaining plates were 14.0% and 6.0%, respectively. Within-replicate error (for the technical duplicates) was 3.3%.

Sexing approach 1: Naive plasma testosterone

For naive neonate tortoises, males had higher average testosterone than females (female mean ± SE in pg/mL: 17.9 ± 1.1, n = 44; male: 51.7 ± 13.5, n = 18; Fig. 2A). The ranges of testosterone levels for male and for female neonates overlapped extensively, but a subset of males exhibited testosterone values well above the female range (Fig. 2A). Adopting an equal-error threshold (TE) to separate males from females would result in a substantial error (approximately 22% for each sex; Table 1), with wide confidence intervals on threshold and error rates (Table 2). Adopting the 90% likelihood thresholds (T90%) to exclude ambiguous individuals would result in excluding over half of all tortoises (Table 2; Fig. 2).

Figure 2 Naive testosterone concentrations by sex in three age classes of gopher tortoise (Gopherus polyphemus) collected from wild nests in Georgia in 2018–20.

Testosterone was higher in males than females for (A) neonates ( <7 days old), (B) hatchlings (4–8 months old), and (C) juveniles (∼45 months old). The labelled dashed line is TE, the threshold at which the the percentage of male samples below that value equals the percentage of female samples above it. The dashed line for juveniles indicates a value roughly mid-way between the two distributions, as they are completely separated. The grey box indicates the TM90 and TF90 thresholds, which indicate the values above and below which an individual is >90% likely to be a male or female, respectively. Note that the asterisked testosterone concentration for the last male sample depicted in (B) is truncated to improve visualization of the male-female overlap, with a true value of 1,193 pg/mL.

Male and female naive hatchling (4–8 months old) tortoises were better separated by testosterone levels (Fig. 2B; Table 2), and male levels were much higher than females (female mean ± SE in pg/mL: 27.9 ± 1.7, n = 66; male: 206.0 ± 49.6, n = 25). Adopting the TE would result in ∼10–12% misclassification for males and females in our dataset (Table 2), with 95% CIs in error rates of 3–18% and 5–13% for females and males (Table 2). Using the T90% thresholds would exclude ∼13% of females and ∼24% of males in our dataset (Table 2). The lower “female” T80% was greater than the higher “male” threshold, indicating that the TE had a likelihood of >80% accuracy for each sex.

Table 2 Thresholds and error rates for distinguishing males and females using circulating testosterone concentration (T in pg/mL; 95% confidence intervals in brackets calculated from 1,000 bootstrap resamples of males and females).

Age	Treatment	T M a	TMF errorb	TMM errorc	T E d	TEF errore	TEM errorf	P g	T PF h	T PM i	TPFF %unkj	TPMM %unkk	
Neonate	Naive	18.3 [18.2-25.05]	5.6% [0-38.9]	34.1% [2.3-40.9]	21.0 [19-23]	22.2% [11.1-38.9]	22.7% [11.4-34.1]	80%	20.7 [17.4-26.7]	23 [19.2-37.1]	25.0% [2.3-52.3]	33.3% [5.6-83.3]	
								90%	16.5 [14.5-21.3]	33.6 [22.9-41.2]	54.5% [15.9-75]	66.7% [22.2-88.9]	
95%	13.9 [12.4-18.7]	39.3 [25.2-43.5]	77.3% [29.5-88.6]	72.2% [27.8-88.9]	
Hatchling	Naive	47.3 [33.95-66.3]	5.9% [0-14.77]	11.4% [1.3-21.5]	49.5 [41-55.2]	11.8% [2.9-17.6]	10.1% [5.1-15.23]	80%	68.5 [54.6-93.41]	38.9 [32.1-44]	1.3% [0-8.9]	2.9% [0-11.8]	
								90%	44.9 [37.4-59.91]	56.1 [42.7-62.3]	12.7% [2.5-25.3]	23.5% [0-35.3]	
95%	33.4 [28.3-47.91]	68.4 [51.5-76.1]	26.6% [8.9-40.5]	29.4% [8.8-41.27]	
Neonate	postFSH	48.7 [17.85-82.75]	34.4% [0-56.2]	18.8% [0-50.15]	29.5 [21.5-51.61]	28.1% [15.6-43.8]	31.2% [18.64-43.8]	80%	15.6 [10.5-46.5]	70.8 [26.09-82.6]	68.8% [6.2-100]	62.5% [18.8-81.2]	
								90%	10.5 [10.5-44.3]	81.3 [36.79-86.9]	93.8% [12.5-100]	65.6% [25-84.4]	
95%	10.5 [10.5-44.3]	86.1 [43.99-90.2]	93.8% [12.5-100]	68.8% [31.2-84.4]	
Hatchling	postFSH	149.9 [105.25-175.45]	4.3% [0-5.29]	1.5% [0-9.2]	127.9 [110.4-171.9]	4.3% [0-8.7]	4.6% [0-9.2]	80%	215.8 [171.73-321.1]	73.6 [61.1-85.6]	0.0% [0-0]	0.0% [0-0]	
								90%	156.4 [127.3-217.8]	107.9 [89.0-122.52]	1.5% [0-7.7]	0.0% [0-8.7]	
95%	123.6 [100.1-169.9]	132.9 [109.4-151.0]	6.2% [0-12.6]	4.3% [0-13.0]	
Notes.

a Testosterone threshold that minimizes total error.

b,c Estimated error rates for females and males using TM.

d Testosterone threshold that has an equal error rate for females and males.

e,f Estimated error rates for females and males using TE.

g Relative likelihood value for calculating TPF and TPM.

h The testosterone threshold below which an individual has P likelihood of being female.

i The testosterone threshold above which an individual has P likelihood of being male.

j,k The percentages of females and males that would be scored as “unknown” sex when adopting TPF and TPM.

Testosterone levels completely separated male and female juveniles (∼3.5 years old; female mean ± SE in pg/mL : 79.8 ± 7.8, n = 11; male: 3,341 ± 432, n = 12; Fig. 2C). The maximum observed female testosterone concentration was 121 pg/mL, while the minimum male concentration was 1,230 pg/mL. A threshold of 700 pg/mL approximates the midpoint between the two distributions (Fig. 2C).

Sexing approach 2: Testosterone following FSH challenge

In our preliminary trials of three different doses of FSH administered to hatchlings from 2018, visual inspection of the results revealed that all three dose levels increased testosterone in males and females (Fig. 3). We did not perform statistics, as there was an inadvertent association of dose with sex (sexes were not known at time of challenge trials), and the sample size within each sex and dose group was small. Nonetheless, it was clear that the lowest dose (0.01 units) effectively elevated testosterone, and that testosterone levels generally returned to their pre-challenge levels after 6 days (Fig. 3), making an FSH challenge using the lowest dose suitable for temporarily stimulating testosterone production. Injections of saline did not elevate testosterone, indicating that FSH itself is responsible for the change in testosterone levels (Fig. S3).

Figure 3 FSH-mediated change in testosterone concentration by dose and sex in young gopher tortoises (Gopherus polyphemus) collected from wild nests in Georgia in 2018–20.

Preliminary tests with three different doses of FSH suggest the lowest dose (0.01 units) is as effective as higher doses at raising testosterone in hatchling tortoises. Lines connect samples collected from the same individual. Samples taken six days after FSH challenge suggest a return to baseline for most individuals. Note different Y axis scales for all three plots.

Neonates challenged with FSH in 2019 had higher testosterone levels than naive testosterone levels in 2020 neonates (data were not paired, and were collected in different years; Fig. 4A), suggesting that FSH similarly elevates testosterone in neonates. In hatchlings in 2020, samples were taken from the same individuals before and after FSH challenge, and the challenge consistently elevated testosterone levels within individual males and females (Figs. 4B–4C). The average difference between naive and post-FSH levels for males was 392 ± 84 pg/mL (n = 23), representing an average increase of 261%, and for females the difference was 25.1 ± 3.5 pg/mL (n = 65), with an average increase of 104%. Permutation t-tests confirmed that there was a significant difference in both the numeric increase and percent increase between sexes (difference in numeric increase: 95% CI [262–502] pg/mL; p < 0.001; difference in percent increase: 95% CI [85.2–236.5]%; p < 0.001). Thus, FSH differentially elevated testosterone in male hatchlings, relative to females.

Figure 4 FSH differentially elevates male testosterone in young gopher tortoises (Gopherus polyphemus) collected from wild nests in Georgia in 2018–20.

(A) Mean neonatal testosterone concentrations [T] were higher in post-FSH individuals than in naive individuals for both females and males. Data are not paired, and each individual was only measured once. Sample sizes are reported inside of each bar. Error bars depict standard errors. Hatchling males (B) and females (C) significantly increased testosterone after receiving a FSH challenge, compared to a naive measurement prior to the challenge. Note different Y axes in each plot. Black squares and error bars indicate within-group means and SEs. Nmales = 23 individuals, Nfemales = 65.

Comparing the distributions of male and female neonates post-FSH challenge, we observed substantial overlap in male and female testosterone concentrations (Fig. 5A), such that the TE of 29.5 pg/mL resulted in ∼30% errors for each sex in our dataset, with 95% CIs up to 43.8% for both sexes (Table 2). Adopting a T90% would exclude 94% of females and 66% of males, making FSH-challenge in neonates a poor method for determining sex.

Figure 5 Post-FSH testosterone concentrations by sex in two age classes of gopher tortoise.

Testosterone concentration [T] was higher in males than females for (A) neonates (<7 days old), and (B) hatchlings (4–8 months old). The labelled dashed line is TE, the threshold at which the percentage of male samples below that value equals the percentage of female samples above it. The dashed line for juveniles indicates a value roughly mid-way between the two distributions, as they are completely separated. The grey box indicates the TM90 and TF90 thresholds, which indicate the values above and below which an individual is >90% likely to be a male or female, respectively.

In contrast, male and female hatchlings (from 2020 when all were given the same dose of 0.01 units) were better separated after FSH challenge (Fig. 5B), such that adopting a TE of 128 pg/mL would result in <5% error for each sex in our dataset (95% CI errors: 0–11% for females, 1–11% for males; Table 2), allowing high confidence in sex assignment (Table 2). The lower “female” T80% and T90% were greater than the higher “male” threshold, indicating that the TE had a likelihood of >90% accuracy for each sex. The T95% thresholds could be adopted with ∼6% loss of males and females, allowing a greater percentage of individuals to be highly accurately sexed when compared to the naive hatchlings (Table 2).

Sexing approach 3: Morphological measurements

Neonates—The random forest model trained on morphological measurements from 174 neonates weakly predicted sex (out-of-bag (OOB) error = 33.3%, with 29.6% error for females and 38.2% error for males; Table 3). Analyzing the subset of 18 male and 44 female neonates from 2020 for which we also had naive testosterone concentrations, OOB error rate was 24.2%, which was not substantially different from a random forest model trained on the testosterone data alone (OOB error = 27.4%). For post-FSH neonates (n = 16 females + 32 males from 2019), the OOB error for the model including morphological and testosterone data was 8%, substantially better than the OOB error for the testosterone-only model (25%). For this model, testosterone, CW/SH, Pmax and SH were the four most important predictors (Table S1).

Table 3 Random forest model results for predicting sex with morphology and testosterone.

Age	Model	n female	n male	OOB error	Female error	Male error	
Neonate	Morph	98	76	33%	30%	38%	
Neonate	Morph + naive T	44	18	24%	21%	33%	
Neonate	naive T	44	18	34%	30%	44%	
Neonate	Morph + postFSH T	16	32	8%	19%	3%	
Neonate	postFSH T	16	32	25%	31%	22%	
Hatchling	Morph	55	23	19%	16%	26%	
Hatchling	Morph subseta	55	23	19%	16%	26%	
Hatchling	Morph + naive T	55	23	10%	9%	13%	
Hatchling	naive T	55	23	19%	20%	17%	
Hatchling	Morph + postFSH T	54	21	12%	9%	19%	
Hatchling	postFSH T	54	21	4%	4%	5%	
Notes.

a Omits AW, AN, TL and GW measurements.

Bold lines indicate the model with lowest error for each age class.

Hatchlings—The random forest model for 55 female and 23 male hatchlings including only morphological data had an OOB error rate of 19.3%, with 16.3% error for females and 26% error for males (Table 3). Removing the AW, AN, TL and GW measurements (and their volume-corrected versions) resulted in the same model performance, indicating these variables contributed little to sex discrimination. Combining morphological measurements with naive testosterone levels (measured at the hatchling age) gave an OOB error of 10.2%, improving on the 19.2% error rate for testosterone alone. Combining the morphological measurements with post-FSH testosterone levels actually reduced model performance from 4% to 12% OOB error.

Sexing approach 4: Geometric morphometrics

The random forest model for 21 female and 15 male neonates including only geometric morphometrics data had an OOB error rate of 52.8%, with 23.8% error for females and 93.3% error for males.

Discussion

We identified several successful approaches to sex young gopher tortoises without the use of laparoscopic surgery (Table 4). The clearest results come from the juveniles (MCL: 101–170 mm, age ∼3.5 years); sexes are completely separable by baseline testosterone, with wide margin for error (Fig. 2C). This may be useful for studies of captive tortoises of this age/size class, obviating the need for surgery. Given the unnatural rearing conditions, it is unclear how these quantitative results map on to wild tortoises in this size class (which could well be much older); these results would ideally be validated with wild animals before use.

Table 4 Summary of best approaches and leave-one-out cross-validation error rates for different datasets (bold lines indicate the approach with lowest error for each age class).

Age	Dataset	Approach	Threshold	n female	n male	Leave-one- out errora	Female error	Male error	
Neonate	Naive T	TEb	21.0	44	18	24%	25%	22%	
Neonate	postFSH T	TE	29.5	16	32	29%	31%	28%	
Neonate	Morph + postFSH T	RF c	NA	16	32	8%	19%	3%	
Hatchling	Naive T	TE	49.5	79	34	12%	11%	12%	
Hatchling	PostFSH T	T E	127.9	81	36	6%	6%	4%	
Hatchling	Morph + postFSH T	RF	NA	54	21	12%	9%	19%	
Juvenile	Naive T	T E	700	11	12	0%	0%	0%	
Notes.

a Overall error for TE reflects uneven sex ratios of samples used in the dataset.

b TE is the threshold that minimizes the differences in error rates between females and males.

c RF: random forest model.

For hatchling tortoises (MCL: 58–76 mm, age ∼4–8 months), a simple threshold for naive testosterone is ∼90% accurate for both sexes. Administering FSH elevated testosterone differentially in males and permitted more accurate sexing, such that most individuals could be sexed with 95% accuracy. Testosterone was only temporarily elevated (Fig. 3), suggesting the FSH method has only transient effects on subjects, making this an excellent substitute for surgery to accurately sex captive tortoises in this age/size class. If population or nest sex ratio data rather than highly accurate individual sexes are desired, naive testosterone may be sufficient, and researchers could avoid the expense and added complication of FSH injection. If individual sex is essential, then FSH treatment may marginally increase accuracy. A drawback of using hatchling testosterone levels is the need to maintain tortoises in the lab until they reach the appropriate age, but if rearing facilities are available, this method could replace laparoscopy as the method of choice for sexing hatchling gopher tortoises.

Our methods were less accurate for neonates. Baseline and post-FSH testosterone only weakly predicted sex. Unlike in 15 month-old snapping turtle juveniles (Ceballos & Valenzuela, 2011) and Arrau neonates (Lance, Valenzuela & Von Hildebrand, 1992), the boost in neonatal testosterone following FSH injection was not sufficiently male-biased to completely separate males and females. However, the combination of basic morphological data and post-FSH data produced relatively accurate predictions in a random forest model (Table 3). Error rates in this model were higher for females than males, and thus could bias overall sex ratios if not corrected. Furthermore, this model was based on a relatively small sample size (16 males, 32 females), and thus is best interpreted as provisional. We provide the random forest model in supplemental code in case researchers wish to use this model with their own data. Given the substantial variability of turtle morphology across populations (Carretero et al., 2005; Djurakic & Milankov, 2020), future studies in other parts of the range may benefit from validating the morphological model before applying it.

We found that morphology alone weakly predicted sex in hatchling tortoises, in contrast to the findings of Burke et al. (1994), and suggest caution when using morphological measurements alone to sex gopher tortoises. The most informative variables in the 81% accurate random forest model predicting sex in hatchlings were CW/VOL, MCL, VOL, Pmin and SW (Table S1). These did not overlap with the results of Burke et al. (1994) from Florida populations that identified CW, SH, VOL, and CW/SH as the most informative. Geometric morphometric analysis was completely uninformative in our population, in contrast to the success this method has achieved in other species (Valenzuela et al., 2004; Ceballos & Valenzuela, 2011; Gómez-Saldarriaga, Valenzuela & Ceballos, 2016). This method may have been more successful with a larger sample size, but given likely morphological variation between populations, a biochemical rather than morphometric approach could be more broadly applicable.

Although we show here that testosterone quantification may be useful in sexing young tortoises, for neonates and for hatchlings the differences between sexes are quantitative and quite small, raising the possibility of error resulting from subtle changes in experimental protocol or differences in animal subjects. A better method may involve the detection of a biochemical signal present in only one sex, making the assay qualitative and thus less vulnerable to quantitative variation resulting from methodological differences across studies. Excitingly, a recent novel approach based on anti-mullerian hormone (AMH) shows promise in loggerhead sea turtle (Caretta caretta) and red-eared slider (Trachemys scripta) neonates (Tezak et al., 2020). Unfortunately, in preliminary tests in our laboratory we were unsuccessful at detecting AMH using similar methods with plasma from male gopher tortoise or red-eared slider hatchlings. We hope future work can expand the development of this method to tortoises and other TSD species. Finally, while considering new methods, it’s important to acknowledge that although laparoscopy requires specialized equipment and skills, it does have several advantages over bioassay-based approaches: it requires no blood draw or sample preservation, results are immediate, it is portable and can be performed internationally where sample collection and transport may be logistically challenging (e.g., Rakotonanahary, Kuchling & Routh, 2015; Emmel et al., 2021), it does not require a laboratory set up for performing hormone assays, and has low cost after the necessary surgical equipment is acquired.

Conclusions

The methods presented here offer simpler and less invasive alternatives to laparoscopy for sexing young gopher tortoises. Neonates may be sexed somewhat accurately using both FSH-challenged testosterone levels and morphology, captive hatchlings may be sexed accurately using naive and FSH-challenged testosterone levels, and captive juveniles of the appropriate size may be easily sexed using naive testosterone levels.

Supplemental Information

Supplemental Information 1 Raw datasets (hormone and morphometric) and R code used to produce all figures and analyses

Click here for additional data file.

Supplemental Information 2 ARRIVE checklist

Click here for additional data file.

Figure S1 Landmarks used in geometric morphometric analysis

We used tpsDIG to place landmarks on standardized photos of each individual’s (A) top of the carapace (top) and (B) plastron (bottom). Photo credit: Kevin Loope

Click here for additional data file.

Figure S2 Parallelism of dilution series of tortoise plasma and testosterone standards (Testosterone ELISA kit; Caymen Chemical, Ann Arbor, MI)

The two-fold serial dilution of a pool of tortoise plasma (1 to 1/1024; n = 20 adult females) is parallel to a two-fold serial dilution of the testosterone standard (Cayman Biological: 500 pg/uL to 3.9 pg/ul). This parallelism suggests that the ELISA antibodies are targeting the same antigens in each series, and thus that the kit is detecting testosterone in plasma samples. It also suggests a lack of matrix interference by other compounds in the plasma. %B/B_0 = %bound/maximum bound

Click here for additional data file.

Figure S3 Saline injections (without FSH) do not elevate testosterone in hatchling gopher tortoises

Click here for additional data file.

Table S1 Importance values for testosterone and morphological measurements in the neonate post-FSH testosterone + morphology random forest model, ordered by mean decrease in accuracy

Click here for additional data file.

We thank Craig Banks, Hailey Baker, Nicole DeSha, Garrett Lawson and Quinton Johnson for help with tortoise care and morphological measurements, and Jack Christie, Abbie Dwire and Curt Vandenberg for assistance in finding nests. Thanks to Roy King, Larry Carlile, and Sheryl McMillan for assistance in site access. Thanks to Kristina Drake and three additional reviewers for providing helpful comments on the manuscript.

Additional Information and Declarations

Competing Interests

Author Contributions

Animal Ethics

Field Study Permissions

Data Availability

1 Any use of trade, firm, or product names is for descriptive purposes only and does not imply endorsement by the U.S. Government.

The authors declare there are no competing interests.

Kevin J. Loope conceived and designed the experiments, performed the experiments, analyzed the data, prepared figures and/or tables, authored or reviewed drafts of the article, and approved the final draft.

David C. Rostal conceived and designed the experiments, performed the experiments, authored or reviewed drafts of the article, and approved the final draft.

M.A. Walden conceived and designed the experiments, analyzed the data, authored or reviewed drafts of the article, and approved the final draft.

Kevin T. Shoemaker conceived and designed the experiments, analyzed the data, authored or reviewed drafts of the article, and approved the final draft.

Elizabeth A. Hunter conceived and designed the experiments, performed the experiments, analyzed the data, authored or reviewed drafts of the article, and approved the final draft.

The following information was supplied relating to ethical approvals (i.e., approving body and any reference numbers):

The Institutional Animal Care and Use Committee of Georgia Southern University approved this research.

The following information was supplied relating to field study approvals (i.e., approving body and any reference numbers):

Field experiments were approved by the Georgia Department of Natural Resources

The following information was supplied regarding data availability:

The raw testosterone and raw morphology data and Code for hormone figures and analyses and morphometric analyses are available in the Supplemental Files.

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
