# Peer review of "A comparison of non-surgical methods for sexing young gopher tortoises (Gopherus polyphemus)"

_PeerJ, doi:10.7717/peerj.13599_

## Round 0.1 · original submission · Minor Revisions

This is a well-conducted and important study that is advancing tortoise conservation. However, there are a few minor changes needed before it can be accepted for publication. These changes can be found in the reviewers’ comments and one annotated pdf. Thank you for this submission!

·

Basic reporting

Overall fantastic, my only recommended additions would be a citation for adding lidocaine to the rostal et al 1994 methodology (Hernandez-Divers SJ, Stahl SJ, Farrell R. An endoscopic method for identifying sex of hatchling Chinese box turtles and comparison of general versus local anesthesia for coelioscopy. JAVMA. 2009;234(6): 800–804; Rakotonanahary TF, Kuchling G, Routh A. In-country anaesthesia for endoscopic sexing of the ploughshare tortoise (Astrochelysyniphora). Solitaire. 2015;26:1-14; emmel et al; Evan S. Emmel, Samuel Rivera, Freddy Cabrera, Stephen Blake, and Sharon L. Deem "FIELD ANESTHESIA AND GONADAL MORPHOLOGY OF IMMATURE WESTERN SANTA CRUZ TORTOISES (CHELONOIDIS PORTERI)," Journal of Zoo and Wildlife Medicine 51(4), 848-855, (12 January 2021).

I really like the component of the conclusion mentioning practicality of endoscopic sex identification in field/less controlled settings, and you could again reference Rakotonanahary et al 2015 and emmel et al 2021 in this section if references are to be included in conclusions.

Experimental design

No comment

Validity of the findings

No comment

Additional comments

Excellent work, very well-written study with an exciting contribution to the field of chelonian captive breeding and sex identification.

Reviewer 2 ·

Basic reporting

The language used throughout the ms is clear and concise and does an excellent job summarizing the literature and the gap that the study attempts to fill.
While it is unfortunate that there is not an easier way to sex neonates, the extraordinarily high rate of success for hatchlings is very encouraging and should be very useful for researchers working on a variety of GT management practices.

I made some small comments about the tables/figures in the PDF.

Experimental design

Excellent job in the methods providing detail and clarity for a series of complicated protocols.

Validity of the findings

No comment

Annotated reviews are not available for download in order to protect the identity of reviewers who chose to remain anonymous.

Reviewer 3 ·

Basic reporting

The writing is clear and unambiguous. The manuscript is well written. I found a few editorial items that I detail below.

Introduction and background are well referenced and relevant.
Figures are excellent, in high quality and clearly present the data. They are well labelled and described.

Raw data were supplied and I examined them.

On line 263 there is a discontinuity in the sentence

On line 313 the present tense is used when it should be past tense.

One line 468 the first word should be capitalized

Experimental design

The research is within the scope of the journal.
The research questions are well defined, relevant and meaningful.
The complicated design is well explained.
The research is performed to the highest technical and ethical standards
The methods are very well described and there is sufficient detail for the experiments to be replicated.

Validity of the findings

This is a beautiful study! The need for it is clearly stated in the Introduction. While the laparoscopic method is the gold standard for sex determination of freshwater turtles and tortoises, there is a need (well documented in this manuscript) for less invasive methods.This study addresses this need and clearly tests several different approaches. In addition, the methods are tested with tortoises of different ages so that makes the study even more valuable. The results are clear and the findings are of fundamental importance for both the biology and conservation of tortoises. The results are applicable to other species as well.

The underlying data are provided, are robust and statistically sound.

The conclusions are well stated and are linked to the original research question.

Additional comments

I wish I had thought of and done this study. It is a masterful job and a real tribute to both innovative science and science geared to the problems of conservation and climate change.

Well done!

---

## Round 0.2 · accepted · Accept

Nice work addressing the reviewers' questions.